# ESCADA: Efficient Safety and Context Aware Dose Allocation for Precision Medicine

**Ilker Demirel**[*]
Bilkent University
ilkerd@ee.bilkent.edu.tr

**A. Alparslan Celik**
Bilkent University
acelik@ee.bilkent.edu.tr

**Cem Tekin**
Bilkent University
cemtekin@ee.bilkent.edu.tr

## Abstract

Finding an optimal individualized treatment regimen is considered one of the most challenging precision medicine problems. Various patient characteristics influence the response to the treatment, and hence, there is no one-size-fits-all regimen. Moreover, the administration of an unsafe dose during the treatment can have adverse effects on health. Therefore, a treatment model must ensure patient *safety* while *efficiently* optimizing the course of therapy. We study a prevalent medical problem where the treatment aims to keep a physiological variable in a safe range and preferably close to a target level, which we refer to as *leveling*. Such a task may be relevant in numerous other domains as well. We propose ESCADA, a novel and generic multi-armed bandit (MAB) algorithm tailored for the leveling task, to make safe, personalized, and context-aware dose recommendations. We derive high probability upper bounds on its cumulative regret and safety guarantees. Following ESCADA's design, we also describe its Thompson sampling-based counterpart. We discuss why the straightforward adaptations of the classical MAB algorithms such as GP-UCB may not be a good fit for the leveling task. Finally, we make *in silico* experiments on the bolus-insulin dose allocation problem in type-1 diabetes mellitus disease and compare our algorithms against the famous GP-UCB algorithm, the rule-based dose calculators, and a clinician.

## 1 Introduction

Precision medicine aims to provide the best possible treatment on an individual level by considering patient characteristics' variability [3, 30]. Many healthcare problems require keeping a physiological variable (e.g., blood glucose level) in a *safe* range and preferably close to a target level. One such example is electrolyte disorders, common among intensive care unit patients. When the blood sodium level falls below 135 milliequivalents per liter (mEq/L) or goes beyond 145 mEq/L, the patient experiences hypo-/hyper-natremia with adverse effects on health [24]. Therefore, correct dosing of electrolytes is crucial to ensure patient safety, and there is no consensus on how to assess the correct dosage for different patient characteristics. Another critical problem is blood pressure disorder. These are hypo-/hyper-tension events where the blood pressure deviates from its standard value and needs to be corrected. Patient characteristics play an essential role in determining the blood pressure response to the therapeutic agent, and they should be taken into account in the dosing process [33].

**Related work and background**    A fair amount of research is dedicated to adaptive clinical trials which aim to identify a drug's effectiveness within a group, including a tradeoff between efficacy and

---

[*]Now a graduate student at MIT CSAIL. Email: demirel@mit.edu

36th Conference on Neural Information Processing Systems (NeurIPS 2022).

toxicity [26, 27, 40, 53]. The algorithms proposed in these works are not applicable to the problem structure considered here for two main reasons. First, the therapeutic agent is not necessarily *toxic*, and our aim is not to maximize the response to the agent but to keep it close to a target level. Therefore, classical upper confidence bound (UCB) based algorithms such as UCB1 [4] or GP-UCB [41] are not applicable for our objective. That is simply because the UCB-based algorithms leverage the *optimism in the face of uncertainty* (OFU) principle to form optimistic estimates of arm outcomes and pick the arm with the highest estimated outcome. However, in our case, *optimism* refers to the proximity of an arm's outcome to the target. This fundamental difference in our task necessitates a novel acquisition strategy. One could simply form pseudo-rewards to maximize, such as $r(n) = -|o(n) - K|$, where $o(n)$ is the outcome at the end of round $n$ and $K$ is the target level. We particularly refrain from doing so as different reasonable choices for the pseudo-reward will lead the algorithm to operate differently in practice. Therefore, we keep the objective (i.e., minimize $|o(n) - K|$) in the most generic form and propose a suitable acquisition strategy instead. We provide more details on our objective and motivation behind designing a new acquisition strategy in §2. Secondly, our goal is to provide personalized recommendations rather than for a group of patients. We approach the safe dose allocation problem from a contextual multi-armed bandit (MAB) [29] perspective with additional safety constraints and propose a novel acquisition function tailored for this problem structure in §3.

To render our acquisition method safe, we propose a safe exploration strategy. There is a surge of interest in safe exploration for Bayesian optimization (BO), Markov decision processes, MABs, and reinforcement learning in general. [15, 17, 31, 54]. [1] propose the linear Thompson sampling (LTS) algorithm for the linear stochastic bandit (LSB) setting by adding a random perturbation to the regularized least-squares estimates of the parameters in a way that the OFU principle can be used. [32] modifies the LTS' randomization procedure to continue leveraging the OFU principle in the face of additional safety constraints and matches LTS' order of regret. [21] proposes a safe algorithm incurring a near-optimal expected regret for the LSB problem as well, which uses the arm outcomes' lower confidence bounds to guarantee the safety of exploration and greedily exploit when it is safe.

There is a strand of literature on "risk-averse" MABs, where the learner is concerned not only with maximizing long-term earnings but also with reducing a certain measure of *risk* [9, 37]. [37, 49, 50] investigate the MAB problem using two risk measures, Mean-Variance and Value-at-Risk, which are widely adopted in financial portfolio management [42]. [8, 14, 20] study the Conditional-Value-at-Risk measure, which captures the tail-risk better compared to the Value-at-Risk measure. Another related area is the "conservative" bandits, where the learner's cumulative reward must always exceed a predetermined fraction of a baseline's [19, 56]. These works, however, do not address *stagewise* safety constraints on instantaneous arm outcomes, which must be explicitly satisfied at any given time.

We operate in a BO framework where we model the objective function as a sample from a Gaussian process (GP). [15, 17] consider BO with stagewise safety constraints. However, they aim to find optimal safe solutions and allow unsafe evaluations during exploration. [2] propose a safe variant of GP-UCB, which employs a pure exploration phase at the beginning and provides upper bounds on its cumulative regret. SafeOPT and StageOPT algorithms provide guarantees on the safety of the exploration process [44, 45]. However, they model the exploration of the safe set as a proxy objective which leads to unnecessary suboptimal evaluations at the boundaries of the safe set [48]. Moreover, they do not provide formal regret bounds. Goal-oriented Safe Expansion (GoOSE) algorithm works with any acquisition function as a *plug-in* safety mechanism and encourages the expansion of the safe set only when necessary [48]. When the query is not guaranteed to be safe, only then GoOSE expands the safe set by evaluating the function at safe points to learn more about the initial query's safety. However, such re-evaluations are not possible within the framework of dynamic treatment regimes since this setup does not allow the administration of multiple doses. Moreover, all the works above consider a one-sided safety constraint ($f(x) \geq c$), whereas we consider a two-sided one as the aim is to keep $f(x)$ in a range ($c_1 \leq f(x) \leq c_2$). We provide a table comparing our work to some existing literature on safe exploration with GPs in Appendix A. Our key contributions are as follows.

We study an important and overlooked problem in medicine that which is relevant in other domains as well, such as demand-side management [7]. We formalize the problem from a contextual MAB perspective via a suitable definition of *regret* as the proxy performance metric in §2. Since our objective is to keep the outcomes close to a target rather than maximize them as in the usual MAB setting, we propose a novel acquisition method in §3. We design a safe exploration scheme for our acquisition function in §3 and derive high probability upper bounds on its regret with safety guarantees in §4. We make *in silico* experiments on type-1 diabetes mellitus (T1DM) disease in

§5. T1DM is characterized by insulin deficiency due to pancreatic $\beta$-cell loss, and it can have adverse effects which might result in hospitalization and death [6]. Therefore, T1DM patients must regulate their blood glucose by administering bolus insulin doses before meals. We optimize the dose recommendation process via *safely* and *efficiently* learning to recommend better doses.

## 2   Problem statement

We denote by $[N]$ the set $\{1, \ldots, N\}$, $\boldsymbol{z} \in \mathcal{Z}$ a context, and $d \in \mathcal{D}$ a dose, where both $\mathcal{Z}$ and $\mathcal{D}$ are compact and convex, and $\mathcal{D} = [0, \overline{D}]$. Let $f : \mathcal{Z} \times \mathcal{D} \rightarrow \Omega$ be the *unknown* function that maps $(\boldsymbol{z}, d)$ pairs to the physiological variable of interest, where $\Omega = [0, \overline{T}]$. At round $n \in [N]$, the learner observes a context, $\boldsymbol{z}_n$, and recommends a dose, $d_n$, to obtain a noisy evaluation of $f$ at $(\boldsymbol{z}_n, d_n)$, given as $y_n = f(\boldsymbol{z}_n, d_n) + \nu_n$, where $\nu_n$ are zero-mean i.i.d. Gaussian with known variance $\sigma^2$. The learner's objective is to keep the physiological variable, $f(\boldsymbol{z}_n, d_n)$, within a safe range and close to the target level. We formalize this objective as a contextual MAB problem with safety constraints as,

$$\text{minimize} \quad R_N = \sum_{n=1}^{N} |f(\boldsymbol{z}_n, d_n) - T| \tag{1}$$

$$\text{subject to} \quad T_{\min} \leq f(\boldsymbol{z}_n, d_n) \leq T_{\max}, \quad \forall n \in [N], \tag{2}$$

where $T_{\min}$ and $T_{\max}$ denote the lower and upper safety thresholds for $f$, respectively, and $T \in (T_{\min} + \alpha, T_{\max} - \alpha)$ is the target value, where $\alpha > 0$. We introduce the non-zero $\alpha$ term to ensure that the target level is not exactly equal to the safety thresholds, which is required later in the analyses. We assume $\forall \boldsymbol{z} \in \mathcal{Z}$, there exists $d_{\boldsymbol{z}}^* \in \mathcal{D}$ such that $f(\boldsymbol{z}, d_{\boldsymbol{z}}^*) = T$.

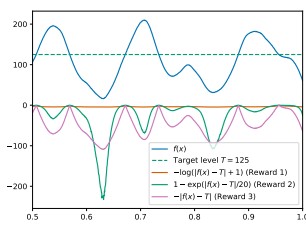

Figure 1: A hypothetical objective function $f(x)$, and three candidate reward functions.

**Regularity assumptions**   Our safe exploration strategy relies on expanding around an initial safe set by exploiting the smoothness properties of the objective function $f(x)$. Without an initial safe set, and some regularity assumptions on $f(x)$, it is not possible to make inferences on the safety of the prospective recommendations [44]. Let $\mathcal{X} = \mathcal{Z} \times \mathcal{D}$ denote the space of all context-dose pairs. Let $k(\cdot, \cdot)$ be a positive definite kernel function on $\mathcal{X}$. We assume that $f(x)$ is a function from the *Reproducing Kernel Hilbert Space* (RKHS) corresponding to $k(\cdot, \cdot)$. In addition, we assume that $f(x)$ has bounded norm in this particular RKHS, i.e., $\|f\|_k < B_f$ [39]. This mild assumption makes $f(x)$ smooth enough to be efficiently learnable by a GP. More precisely, $f(x)$ is $L$-Lipschitz continuous w.r.t. kernel metric $q(\boldsymbol{x}, \boldsymbol{x}') = \sqrt{k(\boldsymbol{x}, \boldsymbol{x}) - 2k(\boldsymbol{x}, \boldsymbol{x}') + k(\boldsymbol{x}', \boldsymbol{x}')}$, where $L = B_f$ [43]. Also, we denote by $q_{\boldsymbol{z}}(d, d') := q((\boldsymbol{z}, d), (\boldsymbol{z}, d'))$. At this point, we define a discretization of $\mathcal{D}$ for every $\boldsymbol{z} \in \mathcal{Z}$ as,

$$\overline{\mathcal{D}}_{\boldsymbol{z}} := \{d_i(\boldsymbol{z}) \in \mathcal{D} \mid i \in \{1, \ldots, k\}\},$$

where $d_1(\boldsymbol{z}) = 0$, $d_i(\boldsymbol{z}) > d_j(\boldsymbol{z})$ for $i > j$, $q_{\boldsymbol{z}}(d_i, d_{i+1}) = \lambda/2L$, $q_{\boldsymbol{z}}(d_k, \overline{D}) < \lambda/2L$, and $\lambda > 0$ is the discretization parameter. We assume that an initial safe set of discretized doses $S_0(\boldsymbol{z})$ is available for each $\boldsymbol{z} \in \mathcal{Z}$. These assumptions allow us to use Gaussian processes (GP) to design our algorithm, and analyze its regret and safety guarantees [36]. A GP is a distribution over functions which is characterized by its mean, $\mu(\cdot)$, and covariance, $k(\cdot, \cdot)$, functions. Once we assume a GP prior over $f(x)$, after observing a set of noisy evaluations $\boldsymbol{y}_N = [y_1 \ldots y_N]^T$ at $A_N = \{\boldsymbol{x}_1, \ldots, \boldsymbol{x}_N\}$, the posterior over $f(x)$ is a GP again with the following mean and covariance functions,

$$k_N(\boldsymbol{x}, \boldsymbol{x}') = k(\boldsymbol{x}, \boldsymbol{x}') - \boldsymbol{k}_N(\boldsymbol{x})^T \left(\boldsymbol{K}_N + \sigma^2 \boldsymbol{I}\right)^{-1} \boldsymbol{k}_N(\boldsymbol{x}')$$
$$\sigma_N^2(\boldsymbol{x}) = k_N(\boldsymbol{x}, \boldsymbol{x})$$
$$\mu_N(\boldsymbol{x}) = \boldsymbol{k}_N(\boldsymbol{x})^T \left(\boldsymbol{K}_N + \sigma^2 \boldsymbol{I}\right)^{-1} \boldsymbol{y}_N,$$

where $\boldsymbol{k}_N(\boldsymbol{x}) = [k(\boldsymbol{x}_1, \boldsymbol{x}), \ldots, k(\boldsymbol{x}_N, \boldsymbol{x})]^T$ and $\boldsymbol{K}_N$ is the positive definite kernel matrix $[k(\boldsymbol{x}, \boldsymbol{x}')]_{\boldsymbol{x}, \boldsymbol{x}' \in A_N}$.

**Comparison with GP-UCB**   Our objective is to keep $f(x)$ close to a target level $T$. As we discussed in §1, one could use the GP-UCB algorithm in [41] if the objective was to *maximize* $f(x)$. In our case, however, we have to define *pseudo-rewards* to maximize such as $-|f(\boldsymbol{z}_n, d_n) - T|$ that are decreasing

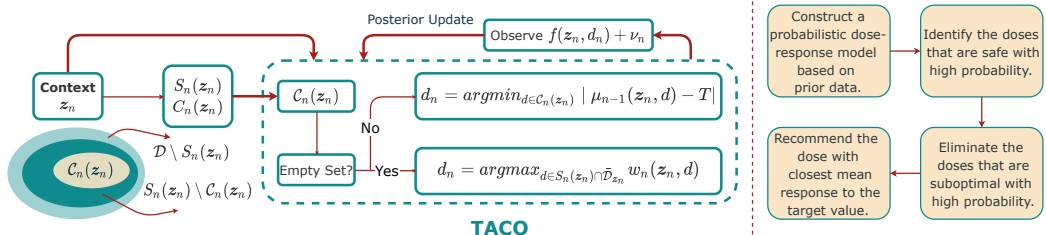

Figure 2: *ESCADA Algorithm Description (left).* Upon observing a context $z_n$ in round $n$, TACO forms the set $\mathcal{C}_n(z_n) \subseteq S_n(z_n)$ after eliminating the doses that are suboptimal with high probability (TACO uses $\mathcal{D}_n = S_n(z_n)$ in Algorithm 1 to ensure safety with high probability). If $\mathcal{C}_n(z_n) \neq \emptyset$, it recommends the dose whose mean response is closest to the target $T$. If $\mathcal{C}_n(z_n) = \emptyset$, it recommends the dose with the widest confidence interval in $S_n(z_n) \cap \overline{\mathcal{D}}_{z_n}$. *Flowchart (right).* A simple interpretation of the dose allocation process intended for domain experts.

with $|f(z_n, d_n) - T|$ to capture the "leveling" task in (1). We present three such reward functions in Figure 1. However, $f(x)$ being smooth and efficiently learnable by a GP does not imply that a reward functional defined on $f(x)$ will be as well. Figure 1 shows that different reward functions can have significantly different landscapes. For instance, it is almost impossible to achieve our task efficiently by using the so-called "plausible" Reward 1. In §5, we compare our algorithms' performances against the GP-UCB's for three reward functions in Figure 1. Also, when $T \neq (T_{\min} + T_{\max})/2$ (which may well be the case, see §5), the reward-based GP-UCB method needs another GP to directly learn $f(x)$ to efficiently satisfy the safety requirements, doubling the computational complexity compared to our algorithms which use a single GP for everything. Finally, by learning $f(x)$ with a GP, we can provide interpretations for our model's recommendations (see Figure 2).

## 3 ESCADA algorithm

We propose ESCADA: Efficient Safety and Context Aware Dose Allocation algorithm. It consists of two blocks: (**i**) an acquisition function, which we call TACO: TArget-based COnfident-acquisition, (**ii**) a safety mechanism to render TACO safe. Algorithm 1 and Figure 2 summarize ES-CADA's design. ESCADA's recommendation procedure can be interpreted to domain experts via the flowchart in Figure 2 as opposed to black-box models [58].[2]

**Acquisition strategy** We propose TACO, a novel acquisition method specifically tailored for the "leveling" task described in §2. At each round $n$, TACO uses the confidence bounds of doses $d \in \mathcal{D}$ for $z_n$ derived from the GP prior as $l_n(z_n, d) = \mu_{n-1}(z_n, d) - \beta_n^{1/2}\sigma_{n-1}(z_n, d)$, and $u_n(z_n, d) = \mu_{n-1}(z_n, d) + \beta_n^{1/2}\sigma_{n-1}(z_n, d)$. We define $\beta_n$ later in a way that the confidence intervals contain the true value of $f$ with high probability (see Lemma 1). Then, using Lipschitz continuity of $f$, we form the final lower and upper confidence bounds for every $d \in \mathcal{D}$ as,

---

**Algorithm 1** ESCADA algorithm

  **for** $n = 1, 2, \ldots$ **do**
    Observe $z_n$ and form $C_n(z_n)$
    Update $S_n(z_n)$ via (3)
    $d_n \leftarrow$ TACO$(C_n(z_n), S_n(z_n))$
    Observe $y_n = f(z_n, d_n) + \nu_n$
    Update GP posterior

---

**Subroutine:** TACO
**Inputs**: $C_n(z_n); \mathcal{D}_n$
$\mathcal{C}_n = \{d \in \mathcal{D}_n \mid T \in C_n(z_n, d)\}$
**if** $\mathcal{C}_n \neq \emptyset$ **then**
  $d \leftarrow \underset{d' \in \mathcal{C}_n}{\arg\min} \ |\mu_{n-1}(z_n, d') - T|$
**else**
  $d \leftarrow \underset{d' \in \mathcal{D}_n \cap \overline{\mathcal{D}}_{z_n}}{\arg\max} \ w_n(z_n, d')$
**return** $d$

---

$$\bar{l}_n(z_n, d) = \max\{l_n(z_n, d), l_n(z_n, d') - Lq_{z_n}(d, d')\}$$
$$\bar{u}_n(z_n, d) = \min\{u_n(z_n, d), u_n(z_n, d') + Lq_{z_n}(d, d')\} ,$$

where $d' = \arg\min_{\hat{d} \in \overline{\mathcal{D}}_{z_n}} q_{z_n}(d, \hat{d})$. We denote by $C_n(z_n, d) = [\bar{l}_n(z_n, d), \bar{u}_n(z_n, d)]$ the confidence interval of a dose $d \in \mathcal{D}$ in round $n$, and by $C_n(z_n) = \{C_n(z_n, d)\}_{d \in \mathcal{D}}$. Finally, we form the confidence widths for each dose $d \in \mathcal{D}$ as $w_n(z_n, d) = \bar{u}_n(z_n, d) - \bar{l}_n(z_n, d)$.

TACO queries a recommendation from a dose set $\mathcal{D}_n$ at each round $n$ upon observing the context $z_n$ in three steps: (**i**): Identify the dose set $\mathcal{C}_n \subseteq \mathcal{D}_n$ whose elements' confidence intervals contain

---

the target value, $T$. **(ii)** If $C_n \neq \emptyset$, recommend the dose in $C_n$ with the closest mean response to the target value $T$. **(iii)** If $C_n = \emptyset$, recommend the dose in $\mathcal{D}_n \cap \overline{\mathcal{D}}_{\boldsymbol{z}_n}$ with the widest confidence interval. In the first step, TACO eliminates the doses which are suboptimal with high probability. This step includes elements of both *exploration* and *exploitation*. A dose whose mean response is close to the target value can be selected (exploitation). On the other hand, if a dose is under-explored, it will have a wider confidence interval which may contain the target, and it stands a chance to be selected (exploration). In the third step, TACO focuses on pure exploration to identify the doses that may be optimal. TACO is *efficient* in the sense that it treats exploration as a proxy objective –in the third step– only when all the feasible doses (i.e., safe) are suboptimal with high probability.

[2]Flowchart assumes that GP-induced confidence intervals are correct, i.e., the event $\mathcal{E}$ in Lemma 1 holds.

**Safety awareness**    We design a safe exploration scheme inspired from the previous works on safe GP optimization [44, 45]. We denote the safe set at round $n$ for the context $\boldsymbol{z}_n$ by $S_n(\boldsymbol{z}_n)$. Let us denote by $\hat{l}_n(\boldsymbol{z}_n, d, d') := \bar{l}_n(\boldsymbol{z}_n, d) - Lq_{\boldsymbol{z}_n}(d, d')$, and $\hat{u}_n(\boldsymbol{z}_n, d, d') := \bar{u}_n(\boldsymbol{z}_n, d) + Lq_{\boldsymbol{z}_n}(d, d')$. We implement the following expansion rule to derive $S_n(\boldsymbol{z}_n)$ each round,

$$S_n(\boldsymbol{z}_n) = S_{n-1}(\boldsymbol{z}_n) \cup \left( \bigcup_{d \in S_{n-1}(\boldsymbol{z}_n)} \{d' \in \mathcal{D} \mid \hat{l}_n(\boldsymbol{z}_n, d, d') \geq T_{\min} \wedge \hat{u}_n(\boldsymbol{z}_n, d, d') \leq T_{\max}\} \right), \quad (3)$$

To satisfy the safety requirements, TACO recommends a dose from $\mathcal{D}_n = S_n(\boldsymbol{z}_n)$ at each round $n$. $S_n(\boldsymbol{z}_n)$ only contains the doses for which $f$ resides in the target interval almost certainly (see Theorem 1). We also define the $\epsilon$-reachability operator $\mathcal{R}_\epsilon$, where $\epsilon > 0$ accounts for the uncertainty in measurements as in [44],

$$\mathcal{R}_\epsilon(S_0(\boldsymbol{z})) := S_0(\boldsymbol{z}) \cup \{d \in \mathcal{D} \mid \exists d' \in S_0(\boldsymbol{z}), \ f(\boldsymbol{z}, d') - Lq_{\boldsymbol{z}}(d, d') - \epsilon \geq T_{\min}$$
$$\wedge f(\boldsymbol{z}, d') + Lq_{\boldsymbol{z}}(d, d') + \epsilon \leq T_{\max}\} . \quad (4)$$

We denote by $\mathcal{R}_\epsilon^n$ the $n$-time reachability operator, which calls $\mathcal{R}_\epsilon$ $n$ times using the previous step's output. Then, $\lim_{n \to +\infty} \mathcal{R}_\epsilon^n(S_0(\boldsymbol{z}))$ represents the subset of $\mathcal{D}$ that can be identified as safe for the context $\boldsymbol{z}$ using the initial safe set $S_0(\boldsymbol{z})$, by observing $f$ up to a statistical certainty restricted by $\epsilon$.

## 4    Theoretical analyses

Consider a sequence of patient contexts $\bar{z} = [\boldsymbol{z}_1 \ldots \boldsymbol{z}_N]$. Let $\mathbb{X}_N = X_1 \times \ldots X_N$ denote the space of all context-admissible recommendation pairs, where $X_n = \boldsymbol{z}_n \times \mathbb{D}_n$, and $\mathbb{D}_n \subseteq \mathcal{D}$ is the admissible dose space for $\boldsymbol{z}_n$. For a given sequence of context-recommendation set $A$, let $\boldsymbol{y}_A$ denote the $|A|$-dimensional vector containing corresponding noisy evaluations of $f$. The quantity governing our regret bounds after $N$ rounds in this scenario is a volatility-adapted maximum information gain term, $\gamma_N^{vol} = \max_{A \subset \mathbb{X}_N} I(\boldsymbol{y}_A; \boldsymbol{f}_A)$, where $\boldsymbol{f}_A = [f(\boldsymbol{x})]_{\boldsymbol{x} \in A}$ and $I(\boldsymbol{y}_A; \boldsymbol{f}_A)$ is the mutual information between $f$ and observations at points in $A$. In the general setting where there is not a fixed context sequence, we have $\gamma_N = \max_{A \subset \mathcal{X}^N} I(\boldsymbol{y}_A; \boldsymbol{f}_A)$. Note that since $\mathbb{X}_N \subseteq \mathcal{X}^N$, we have $\gamma_N^{vol} \leq \gamma_N$. Explicit bounds on $\gamma_N$ depending on $N$ are studied in the literature [41, 51]. In this section, we first derive a high probability upper bound on the cumulative regret of TACO for a fixed context sequence without safety constraints. Then, we bound the regret of ESCADA in a single context scenario with safety constraints. For the former, we have $\bar{z}_1 = [\boldsymbol{z}_1 \ldots \boldsymbol{z}_N]$, and $\mathbb{D}_n = \mathcal{D}$, and we denote the upper bound on the information gain term (see Lemma 2) by $\gamma_N^{vol1}$. For the latter, we have $\bar{z}_2 = [\boldsymbol{z} \ldots \boldsymbol{z}]$, $\mathbb{D}_n = S_n(\boldsymbol{z})$, and we denote the upper bound on the information gain term by $\gamma_N^{vol2}$. We also prove that every dose recommended by ESCADA is safe with high probability (w.h.p.). Detailed proofs for each result can be found in Appendix D.

First, we mention two standard results. Lemma 1 shows that $f(x)$ is contained in the GP-induced confidence intervals w.h.p. and Lemma 2 expresses the information gain in terms of predictive variances.

**Lemma 1.** *(Theorem 1 in [25]) Pick $\delta \in (0, 1)$, and define $\beta_n = 2L^2 + 300\gamma_n \log^3(n/\delta)$, where $L$ is the Lipschitz constant. Let $\mathcal{E} = \{|\mu_{n-1}(\boldsymbol{x}) - f(\boldsymbol{x})| \leq \beta_n^{1/2} \sigma_{n-1}(\boldsymbol{x}), \forall n \in \mathbb{N}, \forall \boldsymbol{x} \in \mathcal{X}\}$. We have $\mathbb{P}\{\mathcal{E}\} \geq 1 - \delta$.*

**Lemma 2.** *(Lemma 5.3 in [41]) The information gain for the points selected can be expressed in terms of the predictive variances. If $\boldsymbol{f}_N = (f(\boldsymbol{x}_n))$, $I(\boldsymbol{y}_N; \boldsymbol{f}_N) = \frac{1}{2} \sum_{n=1}^N \log(1 + \sigma^{-2}\sigma_{n-1}^2(\boldsymbol{x}_n))$.*

The following theorem provides a safety guarantee for ESCADA under the event $\mathcal{E}$ in Lemma 1. The proof depends on an inductive argument on the safe sets constructed by ESCADA.

**Theorem 1.** *Given that an initial safe dose set $S_0(z)$ is available $\forall z \in \mathcal{Z}$, all doses recommended by ESCADA are safe, that is, $T_{\min} \leq f(z_n, d_n) \leq T_{\max} \ \forall n \in [N]$, with at least $1 - \delta$ probability.*

We proposed a novel acquisition function, TACO, for the *leveling* problem described in §2. Theorem 2 provides an upper bound on the regret of TACO without any safety constraints in place.

**Theorem 2.** *Define $\beta_n$ as in Lemma 1 and let $C := 8 / \log(1 + \sigma^{-2})$. Cumulative regret of TACO for a fixed context sequence is upper-bounded as follows,*

$$\mathbb{P}\{R_N \leq \sqrt{CN\beta_N \gamma_N^{vol1}}\} \geq 1 - \delta \ .$$

Next, we introduce a new concept, *safe path*.

**Definition 1.** (Safe Path) For a fixed context $z \in \mathcal{Z}$, we say that there exists a safe path between two doses $d_1, d_2 \in \mathcal{D}$ if the following is satisfied,

$$\eta(d_1, d_2) = \min \left( \min_{d \in [d_1, d_2]} \left( T_{\max} - \epsilon - f(z, d) \right), \min_{d \in [d_1, d_2]} \left( f(z, d) - T_{\min} - \epsilon \right) \right) > 0 \ , \quad (5)$$

where $\epsilon > 0$ is same as in (4). Definition 1 states that if there exists a safe path between two doses $d_1$ and $d_2$, then there is no dose violating or exactly at the safety constraints between them. That is, $f(d) \in (T_{\min} + \epsilon + \eta(d_1, d_2), T_{\max} - \epsilon - \eta(d_1, d_2))$ for all $d \in [d_1, d_2]$. Next, we give the regret bound for ESCADA, which uses TACO with the safe sets $S_n(z_n)$ in (3). We assume a fixed context scenario and show that the safety constraints result in at most a constant addition to the regret.

**Theorem 3.** *If there exists a safe path between at least one dose $d \in S_0(z)$ and $d_z^*$, and we have $q_z(d_1, d_2) = K(|d_1 - d_2|)$ for some monotonically increasing mapping $K : \mathbb{R}^+ \to \mathbb{R}^+$ and for all $d_1, d_2 \in \mathcal{D}$, then the cumulative regret of ESCADA in a safety constrained single context ($z$) scenario can be upper-bounded by setting the discretization parameter $\lambda < \epsilon$ as follows,*

$$\mathbb{P}\{R_N \leq \sqrt{CN\beta_N \gamma_N^{vol2}} + \overline{T}N_z\} \geq 1 - \delta \ ,$$

*where $N_z \in \mathbb{N}$ is a constant independent of $N$.*

Note that since $f(z, d_z^*) = T$ and $T \in (T_{\min} + \alpha, T_{\max} - \alpha)$, one must ensure that $\alpha > \epsilon$ for the possibility of a safe path between some $d \in S_0(z)$ and $d_z^*$ at the first place.

The assumption that $q_z(d_1, d_2) = K(|d_1 - d_2|)$ for a monotonically increasing mapping $K$ holds in our working example where the blood glucose response to insulin dose can be characterized by the carbohydrate factor (CF) [38, 55]. That is, if we let $L \gg$ CF, then we have $f(z, d_1) - f(z, d_2) \leq L|d_1 - d_2|$ for $d_1, d_2 \in \mathcal{D}$, and $q_z(d_1, d_2) = |d_1 - d_2|$. Moreover, this is the case for a variety of widely used kernel induced distance metrics. For the squared exponential kernel $k(\alpha, \beta) = \exp\left(-\|\alpha - \beta\|^2 / 2\sigma^2\right)$, we have (see §2),

$$q_z(d_1, d_2) = \sqrt{2 - 2\exp\left(-|d_1 - d_2|^2 / \sigma^2\right)} \quad (6)$$

Similar observations follow for other radial-basis function kernels (e.g., Laplacian kernel). Theorems 2 and 3 constitute the non-incremental parts in our analysis as they provide explicit regret guarantees for a novel problem structure and acquisition strategy, both with and without safety constraints for a *compact* and *convex* action set. To generalize the bound in Theorem 3 to mixed context scenarios, one needs to impose further assumptions on the regularity of context arrivals over time. We provide experimental results on mixed context scenarios in §5 and show that the inter-contextual information transfer actually improves the performance as expected.

## 5 Experiments

### 5.1 Experimental setup

Online experimentation in the clinical setting is hazardous and it faces ethical challenges [12, 34, 35, 52]. Previous works on dose-finding clinical trials validate their methods either through synthetic experiments or by using external algorithms to fit a dose-response model to real-world data when

the patient group is homogeneous [5, 26, 40]. Such algorithms are not applicable in our case as they assume a shared dose-response model among patients, whereas we aim to learn *personalized* models. We make *in silico* experiments using the open-source implementation [57] of the U.S. FDA approved University of Virginia (UVA)/PADOVA T1DM simulator [23], which is the most frequently used framework in blood glucose control studies [10, 11, 13, 18, 28, 46, 59, 60]. It comes with 30 virtual patients with different individual characteristics: 10 adults, 10 adolescents, and 10 children. The simulator calculates the postprandial blood glucose (PPBG) response of a patient for (meal event, bolus insulin dose) pairs using differential equations and patient characteristics [23]. In our best effort to evaluate the success and potential of ESCADA as a supplementary tool in the clinical setting and to provide external validation, we also compare its performance against a clinician for five virtual adult patients. Our code is available at https://github.com/Bilkent-CYBORG/ESCADA.

**Performance metrics**    When the PPBG level drops below 70 mg/dl (or exceeds 180 mg/dl), hypoglycemia (hyperglycemia) events occur. Both events may lead to life-threatening conditions [6]. Our primary objective is to recommend insulin doses that keep the patients' PPBG level close to the target BG level (see (1)) while not recommending any insulin dose that triggers hypoglycemia or hyperglycemia events (see (2)). We set the target blood glucose (BG) level to 112.5 mg/dl [22]. We gauge an algorithm's performance by combining its regret, hypoglycemia/hyperglycemia frequencies (error frequencies), and glycemic risk indices. Glycemic risk indices are low blood glycemic index (LBGI) and high blood glycemic index (HBGI), and they characterize the risk of hypoglycemia and hyperglycemia events in the long term, respectively [22]. A well-rounded algorithm should have a low cumulative regret together with small risk index values by *safely* and *efficiently* learning to recommend better insulin doses. Besides, we discuss the competing algorithms' consistency since inexplicable variations in medical therapy are undesirable [47]. Precisely speaking, for a *fixed history*, when we query a recommendation from a consistent algorithm multiple times for the same meal event, it should not change. A meal event is a two-element tuple: (carbohydrate intake, fasting blood glucose). We create different meal events via uniform sampling to create an ensemble of different scenarios. We sample carbohydrate intake for each meal event from [20, 80] g, and fasting blood glucose from [100, 150] mg/dl.

**Single meal event (SME) scenario**    In this part, we recommend insulin doses to a patient for the same meal event, assuming that the patient takes the insulin dose directly before the meal. Simulating this setup is helpful for two reasons: (**i**) it tests the performance of the algorithms in the classical *non-contextual* MAB setting, (**ii**) it provides a simple benchmark to understand the performance metrics and to compare them with the contextual setup later. Our objective is to optimize the PPBG 150 minutes after the meal. We make 15 consecutive dose recommendations for a meal event in a single run. We repeat this experiment with 30 different meal events for all 30 patients.

**Multiple meal events (MME) scenario**    In this part, we recommend insulin doses to a single patient for a sequence of different meal events and use the same 30 meal events created in the SME scenario. We make consecutive recommendations for different meal events in a round-robin fashion and recommend a total of 15 doses for each meal event. Precisely speaking, after making a dose recommendation for a meal event, we make recommendations for the other 29 meal events and observe the PPBGs before making the next recommendation for the same meal event. This setup illustrates that the information gained from a context can assist in making decisions for different contexts. Contextual knowledge transfer enables our algorithm to adapt to intra- and inter-daily variability in meal events.

**Algorithms**    We simulate ESCADA and TACO (i.e., without the safety mechanism). Besides, we propose a Thompson sampling (TS)-based algorithm and its safe version (STS), which operate as follows: TS samples a PPBG function from the posterior GP in each round and recommends the dose that achieves the PPBG closest to the target BG. STS implements the safe exploration strategy in §3 and uses TS as the acquisition function. In the final part, we implement the GP-UCB algorithm in [41] using three different "reward" functions in Figure 1 and compare it to our acquisition functions TACO and TS. We use two versions of dose calculators as baselines, whose details are given below.

**Dose calculators**    Dose calculators are commonly used in diabetes care, as they are transparent and interpretable [55]. We use them to initialize the safe dose set for patient and meal event pairs. A calculator recommends an insulin dose via a simple equation, including carbohydrate intake, fasting blood glucose, and patient-specific parameters. They must be fine-tuned to ensure safety which may be challenging. Even when fine-tuned, they may not include some patient characteristics which can affect PPBG in the calculation rule. *Correction doses* constitute 9% of the patients' daily insulin dose intake due to the calculator's failure [55]. More details about bolus calculators are available in Appendix C.

Table 1: "-TC" indicates that tuned calculator was used. Target PPBG level is $T = 112.5$ mg/dl. "PPBG" column is averaged over observations for all 30 patients, 30 meal events per patient, and 15 recommendations per meal event. "Hyper" and "Hypo" columns denote the hyperglycemia and hypoglycemia event frequencies, respectively, averaged over all 30 patients. Similarly, HBGI and LBGI risk indices are averaged over all 30 patients. We report (`mean ± standard deviation`).

|  | Algorithm | PPBG | Hyper | Hypo | HBGI | LBGI |
|---|---|---|---|---|---|---|
|  | Calc. | $144.0 \pm 39.5$ | $.143 \pm .217$ | $.0614 \pm .189$ | $3.84 \pm 3.41$ | $1.37 \pm 3.95$ |
|  | Tuned Calc. | $123.7 \pm 18.1$ | $0$ | $.0021 \pm .010$ | $0.83 \pm 0.66$ | $0.24 \pm 0.47$ |
| SME | TS | $119.8 \pm 42.2$ | $.046 \pm .029$ | $.0216 \pm .028$ | $1.52 \pm 1.25$ | $1.01 \pm 2.31$ |
|  | TACO | $121.7 \pm 50.4$ | $.049 \pm .032$ | $.0175 \pm .019$ | $1.89 \pm 1.63$ | $0.52 \pm 0.46$ |
|  | STS | $121.6 \pm 24.8$ | $.031 \pm .063$ | $.0029 \pm .010$ | $1.07 \pm 1.33$ | $0.15 \pm 0.23$ |
|  | ESCADA | $122.2 \pm 20.0$ | $.015 \pm .030$ | $.0031 \pm .008$ | $0.77 \pm 0.82$ | $0.11 \pm 0.24$ |
|  | STS-TC | $\mathbf{117.1 \pm 11.9}$ | $\mathbf{.002 \pm .004}$ | $\mathbf{.0004 \pm .001}$ | $\mathbf{0.28 \pm 0.24}$ | $\mathbf{0.05 \pm 0.05}$ |
|  | ESCADA-TC | $\mathbf{116.1 \pm 12.5}$ | $\mathbf{.002 \pm .004}$ | $\mathbf{.0007 \pm .003}$ | $\mathbf{0.26 \pm 0.21}$ | $\mathbf{0.07 \pm 0.09}$ |
| MME | GP-UCB-1 | $124.1 \pm 87.0$ | $.179 \pm .050$ | $.2618 \pm .202$ | $5.43 \pm 2.26$ | $15.4 \pm 22.5$ |
|  | GP-UCB-2 | $103.7 \pm 59.4$ | $.080 \pm .060$ | $.2873 \pm .254$ | $2.21 \pm 1.58$ | $16.0 \pm 27.0$ |
|  | GP-UCB-3 | $111.0 \pm 32.6$ | $.022 \pm .010$ | $.0648 \pm .076$ | $0.73 \pm 0.33$ | $3.45 \pm 5.17$ |
|  | TS | $\mathbf{112.4 \pm 14.4}$ | $\mathbf{.003 \pm .003}$ | $.0107 \pm .011$ | $\mathbf{0.16 \pm 0.18}$ | $0.53 \pm 0.95$ |
|  | TACO | $\mathbf{113.7 \pm 19.2}$ | $\mathbf{.006 \pm .031}$ | $\mathbf{.0010 \pm .002}$ | $\mathbf{0.29 \pm 1.18}$ | $\mathbf{0.07 \pm 0.04}$ |
|  | STS | $116.5 \pm 12.5$ | $\mathbf{.004 \pm .015}$ | $\mathbf{.0007 \pm .002}$ | $\mathbf{0.32 \pm 0.49}$ | $\mathbf{0.05 \pm 0.05}$ |
|  | ESCADA | $116.9 \pm 13.1$ | $\mathbf{.006 \pm .017}$ | $\mathbf{.0005 \pm .002}$ | $\mathbf{0.34 \pm 0.55}$ | $\mathbf{0.04 \pm 0.04}$ |

We consider two setups. First, we use a calculator setting that occasionally fails to provide safe dose recommendations and sacrifice the assumption that an initial safe set, $S_0(z)$, is always available. Then, we use tuned calculators for each patient and ensure that $S_0(z)$ is *almost* always available.

### 5.2 Discussion of results

**Safety** Ensuring patient safety is pivotal. Theorem 1 shows that ESCADA recommends safe doses with high probability when an initial safe dose set is available. However, the initially provided set may not always be safe in reality due to calculator or clinician mistakes. We simulate two scenarios when an initial safe set is almost always available and not. For the latter, Table 1 shows that the error frequencies of ESCADA are not zero. We expect that error since the calculator fails to consistently provide safe doses in the beginning. However, ESCADA yields significantly lower error frequencies and risk index values than the calculator. That improvement stems from ESCADA's ability to gradually identify and recommend safe doses, even when initially misdirected. We plot consecutive dose recommendations by ESCADA in SME scenario for three different meal events in Figure 4. For each of these meal events, rule-based calculator fails to provide safe doses in the beginning. Notwithstanding, ESCADA expands its safe set in the right direction and eventually recommends safe doses. Figure 3 and Table 1 confirm the safety mechanism's effectiveness as ESCADA and STS yield significantly better safety metrics than the unsafe algorithms, TACO and TS, especially for hypoglycemia. Next, we manually tune the calculator parameters for each patient separately so that it successfully provides an initial safe set almost always (Tuned Calc., Table 1). Table 1 shows that ESCADA-TC and STS-TC yield remarkably lower error frequencies and risk indices, along with better PPBG distributions.

**Regret** Minimizing the regret is equivalent to recommending doses that lead to PPBG values close to the target BG by (1). We observe from Figures 3 and 5, and Table 1 that ESCADA(-TC) and STS(-TC) significantly outperforms the (tuned) calculator. Figure 5 shows that TACO and TS incur lower cumulative regrets than ESCADA and STS in the MME scenario. That is a natural trade-off between safety and regret since the safety mechanism restricts the allocation of a dose before it is identified as safe. Therefore, a safe algorithm yields higher regret when the initial safe set is far from the optimal dose.

**Inter-contextual information transfer** We investigate the efficiency of GP-induced smoothness in *transferring* information between different contexts. We mark an evident advancement in PPBG distributions and safety metrics in the MME scenario compared to the SME scenario in Table 1. Examining Figure 6, we observe that ESCADA expands the safe dose set and identifies the optimal dose faster in the MME scenario. Remember that ESCADA recommends doses for different meal

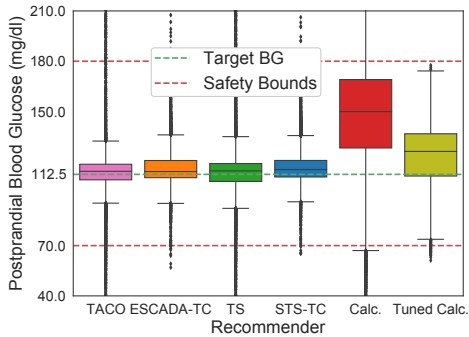

Figure 3: PPBG distribution boxplots in SME scenario. "-TC" suffix indicates that the tuned calculator is used.

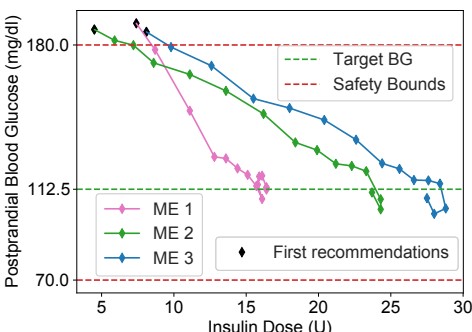

Figure 4: Consecutive dose recommendations to three different meal events with *unsafe* $S_0(\boldsymbol{z})$ in SME scenario.

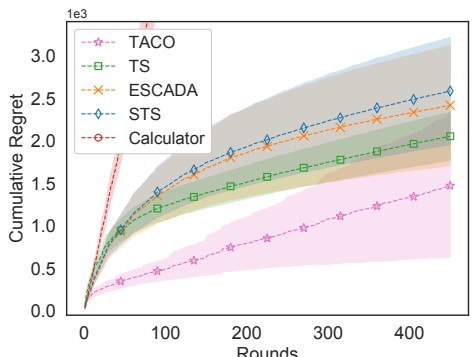

Figure 5: The cumulative regrets averaged over all 30 patients in MME scenario ($\pm$ 0.25 standard deviation).

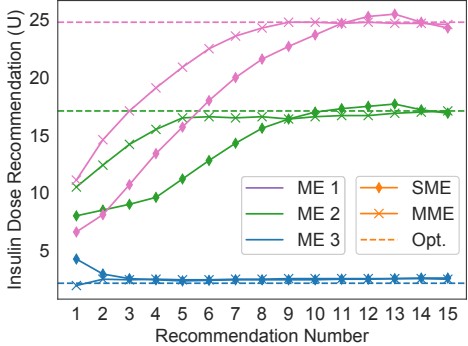

Figure 6: Consecutive dose recommendations for three different meal events (ME) in SME and MME scenarios.

events between two consecutive recommendations for the same meal event in the MME scenario. That is, the information gained from a context improves the performance for other contexts. Besides, we observe significant advances in the safety metrics of TACO and TS in the MME scenario as well.

**Comparison with GP-UCB** We compare our acquisition functions, TACO and TS, against the adaptations of the GP-UCB algorithm as described in §2 for three different reward functions in Figure 1. "GP-UCB-X" uses "Reward X" in Figure 1, which are defined as follows at each round $n$,

$$r_1(n) = -\log(|y_n - T| + 1) \qquad r_2(n) = 1 - \exp(|y_n - T|/20) \qquad r_3(n) = -|y_n - T|$$

We have $y_n$ instead of $f(\boldsymbol{z}_n, d_n)$ as the observations are noisy. Figures 7, 8, and 9 show that GP-UCB's performance varies wildly for different rewards, and it is outperformed by TACO and TS. The practitioner needs to choose a "good" reward function for each problem. Our algorithms do not require that.

**Consistency** Figures 3 and 5, and Table 1 reveal that ESCADA and STS yield similar results. Both algorithms use GPs and have $\mathcal{O}(n^3)$ time and $\mathcal{O}(n^2)$ memory complexities where $n$ is the number of observations. The key difference between them is that STS strikes the balance between exploration and exploitation through intrinsic randomization. That is, for a fixed patient history, STS can make different recommendations for the same meal event in test time, damaging its interpretability and leading to undesired inexplicable variations in the treatment [47]. On the other hand, ESCADA trades-off the exploration and exploitation through the explicit and deterministic machinery described in §3 which makes it a fairly interpretable model. Moreover, even though we design and test STS, we do not provide an upper bound on its regret as opposed to ESCADA, which is an interesting future work.

**Clinician comparison** We compare ESCADA's performance against a clinician's for five virtual patients. For each patient, we provided the clinician with 20 samples in the form of (meal event, insulin dose, PPBG) and asked her to make recommendations for 20 *unseen* meal events. We provided ESCADA with the same 20 samples for each patient and queried recommendations for the same 20 test meal events. Figure 10 shows that the clinician performs slightly worse than the calculator,

and ESCADA outperforms both significantly. These results suggest that making inferences about a patient's dose response is not trivial, and ESCADA is promising supplementary tool in clinical setting. Moreover, ESCADA can provide the clinicians with various useful statistics regarding dose responses, such as the confidence region of the response, hypo-/hyper-glycemia probabilities, or probability of response residing in a specific interval for a given patient, meal event, and insulin dose.

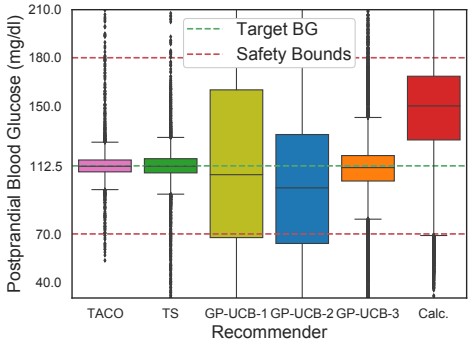

Figure 7: PPBG distribution boxplots for TACO, TS, GP-UCB, and the calculator in MME scenario.

Figure 8: Cumulative regrets for TACO, TS, GP-UCB, and calculator in MME scenario ($\pm$ 0.25 standard deviation)

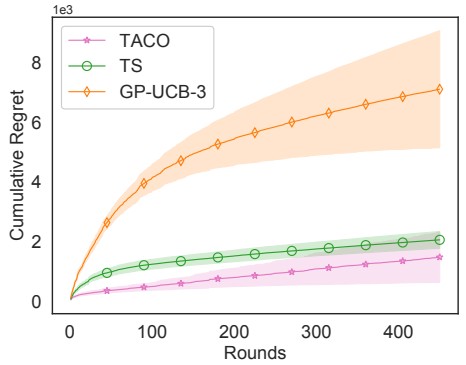

Figure 9: Cumulative regrets for TACO and the best GP-UCB in MME scenario ($\pm$ 0.25 standard deviation).

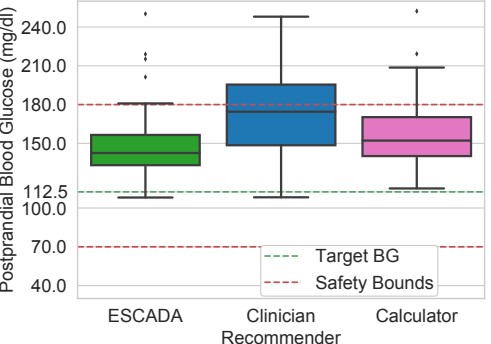

Figure 10: PPBG distribution boxplots for five virtual patients in the clinician comparison experiment.

## 6 Concluding remarks

We formalized and studied a prevalent problem in medicine, *safe leveling*, and proposed TACO, a novel acquisition function tailored for this problem structure. As safety is crucial in healthcare, we proposed a safe exploration strategy to render TACO safe. Combining these two blocks, we proposed ESCADA, a *safe* and *efficient* learning algorithm, and provided safety guarantees and upper bounds on its cumulative regret. Through extensive *in silico* experiments on the bolus-insulin dose allocation problem for type-1 diabetes disease, we showed our algorithms' effectiveness over the rule-based dose calculators and straightforward adaptations of the GP-UCB algorithm for the *safe leveling* task. We also compared ESCADA's performance against a clinician's to provide external validation and discussed its potential as a complementary instrument in clinical settings. ESCADA can also be used in other safety-critical decision-making problems where the goal is to safely control a target variable.

**Acknowledgments:** This work was supported by the Scientific and Technological Research Council of Turkey (TUBITAK) under Grant 215E342. Ilker Demirel was also supported by Vodafone as part of 5G and Beyond Joint Graduate Support Programme coordinated by Information and Communication Technologies Authority. The clinician experiment was done as part of the TUBITAK Project 215E342 under the supervision of a Professor in the Dept. of Internal Diseases in Umraniye Training and Research Hospital, Istanbul, Turkey. Dose recommendations for the virtual patients are provided by a clinical dietician working in the same hospital. Research conducted within the scope of the TUBITAK Project 215E342 has been approved by the ethics committee of the hospital.

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
