# OpenReview forum: "ESCADA: Efficient Safety and Context Aware Dose Allocation for Precision Medicine"
_NeurIPS.cc/2022/Conference — NeurIPS 2022 Accept_

### Official Review · Reviewer_jeRY · 2022-07-11

**Rating:** 4
**Confidence:** 3
**Ethics Flag:** Yes
**Soundness:** 2 fair
**Presentation:** 3 good
**Contribution:** 3 good

**Summary:**

The paper introduces and evaluates a multi-armed bandit (MAB) algorithm for insulin dose allocation related to Type-1 Diabetes (T1D). Its technical contributions in enhancing safety in T1D by preventing hyperglycemia and hypoglycemia by context-aware and personalised dosing of insulin before meals is fairly convincing, based on evaluations on an appropriately chosen simulator.

**Questions:**

Please separate the 10 adults, 10 adolescents, and 10 children in evaluations and consider if the target PPBG level should be adjusted by this age group as well.

Please explain more about the evaluation conducted by the clinician (e.g., the cases analysed, implementation of the assessment task, ethics approval, and informed consent)

**Ethics Review Area:**

["Privacy and Security (e.g., consent)"]

**Limitations:**

Clinical aspects of the study could be clearer. For example, the guidelines by https://www.jmir.org/2016/12/e323/ may help. In particular, I encourage clarifying the rationale, clinical implications, and limitations of the model; see the JMIR paper for further details about these topics.

**Strengths And Weaknesses:**

The paper introduces and evaluates a multi-armed bandit (MAB) algorithm for insulin dose allocation related to Type-1 Diabetes (T1D). Its technical contributions in enhancing safety in T1D by preventing hyperglycemia and hypoglycemia by context-aware and personalised dosing of insulin before meals are fairly convincing, based on evaluations on an appropriately chosen simulator. However, the evaluation should have differentiated the 30 simulated patients instead of presenting averaged results over all these patients, regardless of their age. For instance, glucose control of adolescents is substantially harder than for adults, and hence, I am used to separating adults, adolescents, and children in evaluating algorithms and medical interventions. Similarly, in order to contribute to precision medicine, as argued by the paper, for example, the target PPBG level should have been adjusted by this age group. Finally, I would have wanted to know more about the clinician that was one of the comparisons in the evaluation; I really liked this more human-centric aspect of the evaluation but would like to know more about this experiment so that future studies could follow the same experimental design if they wanted. Also it was unclear how the simulate patient cases to be analysed by this clinician were chosen and how the human judgement and decision-making was implemented in this part experiments. Last but not least, I could not find details about this experiment (e.g., the cases analysed, implementation of the assessment task, etc.) involving a human participant (i.e., the clinician) as a participant, or obtaining the related ethics approval and informed consent, from the paper or its supplement.

---

> ### Author Response · Authors · 2022-07-31
> **Rebuttal**
>
> We thank the reviewer for their thoughtful comments and valuable insights.
>
> ## Patient Subgroups
>
> We set the target blood glucose level and the safety bounds following the guidelines in [6] and [22], same for all patients. However, we agree that the treatment strategies differ between subgroups and patients. From a precision medicine perspective, we intend to show that the proposed algorithms can learn different personalized strategies for each patient that are necessary to achieve the same target blood glucose objective. We kindly ask the reviewer to see the **Appendix B - Additional Experimental Results** in the **revised version of the supplementary material**, where we plot the postprandial blood glucose distributions separately for three patient subgroups, adult, adolescent, and child (Figure 11a, Line 564). We also plot the dose recommendation distributions and show that our algorithms are personalized in the sense that they learn to recommend different doses to different patients to achieve the same target BG level, $T=112.5$ mg/dl.
>
> Moreover, we make experiments where the target blood glucose level $T$ is different for each patient subgroup. Note that, however, our objective is not to find some optimal $T_{\text{min}}$, $T_{\text{max}}$, $T$ values in general, but to learn to provide dose recommendations according to the values determined by the clinician. We kindly ask the reviewer to see Figure 12 and the related discussions (Lines 565-577) in **Appendix B - Additional Experimental Results** in the **revised version of the supplementary material**. Our algorithms continue to optimize the dose recommendations for each patient subgroup with different target blood glucose levels, corroborating the claim that our algorithms can be personalized. The clinician can set the safety bounds $T_{\text{min}}$ and $T_{\text{max}}$, and the target level $T$ separately for each patient depending on the treatment objective, without having to tune any parameters.
>
> ## Clinician Qualifications and Experiments
>
> The clinical experiment is conducted within the scope of an interdisciplinary joint research project and under the supervision of a Professor in the Department of Internal Diseases in a Training and Research Hospital. Dose recommendations for the virtual patients are provided by a clinical dietician working in the same hospital and they are reviewed by the supervisor. As the experiment only involves synthetic patients, and no data is collected from real patients, no ethics approval or informed consent forms were filled. We will acknowledge our clinical collaborators in the final version of the manuscript, which is excluded from the submission to preserve anonymity.
>
> In the clinician experiments, we use five virtual patients. For each patient, we create 20 meal events for training by random sampling without any particular structure similar to other scenarios (see Lines 279-280). Then, we make a random insulin dose recommendation for every meal event and patient pair and record the postprandial blood glucose values after some time (abbreviated as “tmbg” in the .csv files). The training data for the clinician experiments is available in the supplementary material, under the directory: “./ESCADA\_code/experiments/clinician\_comparison/train\_data”. We then ask the clinician to inspect the training data to make inferences about patients’ blood glucose response to insulin. Then, the clinician makes dose recommendations for 20 **unseen** meal events  (test events) for each patient to achieve a close blood glucose level to the target level 150 minutes after the meal (same as the other experiments).
>
> Even though this is a small experiment, we believe it is a nice way to provide external validation and gain insights into the task complexity, especially when real-life experiments are not possible. We will extend the **Comparison against a clinician** part in Section 5 in the final version of the manuscript to provide a more detailed outline of our experimental design, so that future studies can follow a similar procedure with ease.
>
> ## Limitations
>
> We thank the reviewer for sharing a useful resource. We discuss some limitations of our algorithm in the relevant parts of the manuscript (e.g., see Lines 244-247 and 108-113). We will dedicate a separate paragraph in the **Concluding remarks** section to highlight and restate the limitations of our algorithm. Moreover, as we have attempted to do so in this rebuttal, we will further clarify our approach to personalized treatment and the reasoning behind our experimental design, including the clinician experiments.
>
> Finally, we would like to note that outside the T1D problem considered, the manuscript introduces a generic problem, **leveling**, to the multi-armed bandit framework, and proposes a novel algorithm with theoretical safety and performance guarantees that can be directly used for other problems as well, where the objective is to safely control a target variable.

---

> > ### Comment · Reviewer_jeRY · 2022-08-09
> > **Post-Rebuttal Response**
> >
> > I have considered all reviews and all authors' rebuttal responses. In summary, I think they have excelled in addressing most crucial constructive comments given by reviewers and given the importance of the topic and strong support indicated by reviewers, I am supporting the acceptance of this paper to NeurIPS; although it may be somewhat incomplete in certain aspects, it should attract interest and future studies as part of the scientific program and proceedings of NeurIPS.

---

> > > ### Author Response · Authors · 2022-08-09
> > > **thank you Reviewer jeRY**
> > >
> > > We are delighted to hear that you were satisfied with our response! We are also grateful that you took your valuable time to read the other reviews & responses and let us know about your post rebuttal thoughts.
> > >
> > > Best regards,
> > > Authors

---

> ### Author Response · Authors · 2022-08-07
> **Dear Reviewer jeRY**
>
> We would like to extend our gratitude for your valuable comments once again! We hope that our response (31 Jul) has addressed your concerns, and we would be happy to discuss your further comments after reading the response.

---

### Official Review · Reviewer_oCtP · 2022-07-11

**Rating:** 7
**Confidence:** 3
**Soundness:** 4 excellent
**Presentation:** 4 excellent
**Contribution:** 4 excellent

**Summary:**

Consider the safe dose allocation problem in precision medicine. This paper proposes a contextual multi-armed bandit algorithm with the objective to keep the outcomes close to a target level. The proposed algorithm has high probability upper bounds on cummulative regrets and also possess a two-sided safety guarantee.

**Questions:**

- Typo found in the pseudo code of TACO. In the line after 'else', I believe it should be $w_n(\mathbf{z}_n,d')$ instead of $d$.
- $\epsilon$ in definition 1 is unclear.

**Limitations:**

None.

**Strengths And Weaknesses:**

# Strengths
- The paper is well organized
- The objective is an interesting problem. Instead of maximizing the outcome, this paper aim to keep the expected outcome close to a target level.
- The 'TACO' algorithm is novel. Both exploration and exploitation are addressed. The action when all safe doses are sub-optimal is interesting.

#Weaknesses
- The figures in section 5 are too small to read.

---

> ### Author Response · Authors · 2022-07-31
> **Rebuttal**
>
> We thank the reviewer for their thoughtful comments and valuable insights.
>
> We will change the presentation of figures 3-6 and figures 7-10 from 1x4 format to 2x2 format to make them larger in the final version, where an extra page is granted. We will also increase the font sizes in the axis labels and the legends.
>
> We thank the reviewer for pointing out a typo which we will fix in the final version.
>
> The $\epsilon$ term in **Definition 1** is the same as in the definition of the $\epsilon$-reachability operator, $\mathcal{R}_{\epsilon}$ (see Lines 183 - 187). We will explicitly mention this in **Definition 1** to avoid confusion in the final version.
>
> Finally, we thank the reviewer for recognizing the strategy when all the safe doses are sub-optimal. That part of our novel acquisition function TACO is indeed the crux of the analysis in bounding the cumulative regret (makes Lemma 5 in the appendix possible).

---

> ### Author Response · Authors · 2022-08-07
> **Dear Reviewer oCtP**
>
> We would like to extend our gratitude for your valuable comments once again! We hope that our response (31 Jul) has addressed your concerns, and we would be happy to discuss your further comments after reading the response.

---

### Official Review · Reviewer_pdcL · 2022-07-12

**Rating:** 8
**Confidence:** 3
**Soundness:** 4 excellent
**Presentation:** 3 good
**Contribution:** 4 excellent

**Summary:**

The authors investigate a problem that they refer to as leveling. Shortly, it is a prevalent medical problem in which the treatment aims to keep a physiological variable in a safe range and preferably close to a target level.

Their proposed algorithm is a multi-armed bandit-based for the leveling task, which aims to make safe, personalized, and context-aware dose recommendations. As a theoretical contribution, they derive probability upper bounds on its cumulative regret and safety guarantees.

Additionally, they conducted in silico experiments on the bolus-insulin dose allocation problem in type-1 diabetes mellitus disease by comparing their algorithm against the GP-UCB baseline.


**Questions:**

It is unclear if the GP-UCB was also initialized using the same scheme for TACO and TS, i.e., the dose calculators proposed in [55]. Also, it would be interesting to see the effect of using random initialization of all algorithms to understand the impact on the dose calculators through epochs.

**Limitations:**

The authors need to include the limitations of their method in the concluding remarks.


**Strengths And Weaknesses:**

The dose-finding problem is challenging, given the ethical concerns from online experimentation with real patients. This paper uses a simulator to bring experimental results. However, it compares its performance against a clinician for virtual adult patients.

It would be worth including the qualifications of the clinician that evaluate the unseen meal events. Possibly, in the appendix.

As a minor improvement, the authors may increase the font size of figures, such as Figures 3-6.

---

> ### Author Response · Authors · 2022-07-31
> **Rebuttal**
>
> We thank the reviewer for their thoughtful comments and valuable insights.
>
> ## Clinician Qualification
>
> The clinical experiment is conducted within the scope of an interdisciplinary joint research project and under the supervision of a Professor in the Department of Internal Diseases in a Training and Research Hospital. Dose recommendations for the virtual patients are provided by a clinical dietician working in the same hospital and they are reviewed by the supervisor. As the experiment only involves synthetic patients, and no data is collected from real patients, no ethics approval or informed consent forms were filled. We will acknowledge our clinical collaborators in the final version of the manuscript, which is excluded from the submission to preserve anonymity.
>
> ## Initialization Scheme for the Algorithms
>
> TACO and TS are the novel **acquisition strategies** proposed in the manuscript for the **leveling** task. ESCADA and STS use TACO and TS as the acquisition functions, respectively, and they employ an **additional** safety mechanism. The safety mechanism ensures that TACO and TS make recommendations only from a dose set, $S_n(z_n)$, that is safe with high probability. For the safety mechanism to work, we need at least one safe dose for any given context $z \in \mathcal{Z}$ (as we mention in the Theorem statements). In the experiments, we provide the initial safe dose, $S_0 (z)$, to ESCADA and STS using the dose calculator in [55] (both the standard calculator and the manually tuned calculator).
>
> TACO, TS, and GP-UCB do not use any safety mechanism or initial safe doses provided by the dose calculators. While it is possible to design a safe exploration scheme for the GP-UCB acquisition function as well, it may double the computational complexity (see Lines 139 - 142).
>
> In comparing TACO and TS to GP-UCB (and not considering the safety), our intention is to focus on demonstrating the necessity of designing **novel acquisition strategies**  for the **leveling** task rather than using the readily available methods via pseudo-reward definitions (see Lines 131 - 139). TACO, TS, and GP-UCB are initialized with the same Gaussian process (GP) prior. Their dose recommendation strategies, however, are different as we explain in detail.
>
> As for the random initialization of the algorithms, we mentioned that TACO, TS, and GP-UCB are initialized with the same GP prior and do not use initial safe doses. In case of ESCADA, we actually show that when the initial doses are not safe, it still converges towards the safe doses and starts making optimal recommendations (see Figure 4). This result suggests that even if we randomly selected the initial dose set required by the ESCADA and STS algorithms, they would still manage to find their way towards safer and better doses, since STS has similar performance results with ESCADA (see Table 1). However, one would expect the performance to be worse (e.g., slower convergence towards the optimal dose, hence incurring higher regret) for the random initializations of the initial dose set that are far from the safe region and the optimal dose.
>
> ## Presentation of Figures
>
> We will change the presentation of figures 3-6 and figures 7-10 from 1x4 format to 2x2 format to make them larger in the final version, where an extra page is granted. We will also increase the font sizes in the axis labels and the legends.
>
> ## Limitations
>
> We discuss the limitations of our algorithm in the relevant parts of the manuscript (e.g., see Lines 244 - 247, and Lines 108 - 113). We will dedicate a separate paragraph in the **Concluding remarks** section to highlight and restate the limitations of our algorithm.

---

> ### Author Response · Authors · 2022-08-07
> **Dear Reviewer pdcL**
>
> We would like to extend our gratitude for your valuable comments once again! We hope that our response (31 Jul) has addressed your concerns, and we would be happy to discuss your further comments after reading the response.

---

### Official Review · Reviewer_iS7Z · 2022-07-19

**Rating:** 4
**Confidence:** 4
**Soundness:** 2 fair
**Presentation:** 3 good
**Contribution:** 2 fair

**Summary:**

The authors study a prevalent medical problem where the treatment aims to keep a physiological variable in a safe range and preferably close to a target level. They propose ESCADA, a multi-armed bandit algorithm tailored for the above leveling task, to make safe, personalized, and context-aware dose recommendations.

**Questions:**

Questions:

1. In your problem setup, does T depend on z_n in (1)?

2. How was the confidence interval estimated? This piece seems to be missing in the manuscript.

**Limitations:**

Please see the weakness and questions.

**Strengths And Weaknesses:**

Strength:

1. They consider constraints on instantaneous outcomes and propose efficient algorithms to achieve their goal.

2. They provide safety guarantees and upper bounds on cumulative regret.


Weakness:

1. Is \alpha a tuning parameter?  There many too many tuning parameters, which makes the method hard to be used in practice.
How would the choice of Tmin Tmax affect the results of the algorithm?

2. What is the complexity of the algorithm? How does it depend on the cardinality of dose sets?

---

> ### Author Response · Authors · 2022-07-31
> **Rebuttal**
>
> We thank the reviewer for their thoughtful comments and valuable insights.
>
> ## Answer to Weakness 1
>
> $\alpha$, $T$, $T_{\text{min}}$, and $T_{\text{max}}$ are not hyperparameters to be tuned, but clinician inputs. The clinician can easily set a different target level, $T$, and safety threshold values, $T_{\text{min}}$ and $T_{\text{max}}$ for different patients as she best sees fit, and our algorithms will handle the rest.
>
> In our experiments, we want to control the blood glucose level. We use the lower and upper safety thresholds $T_{\text{min}}=70$ mg/dl and $T_{\text{max}}=180$ mg/dl, respectively, and the target level $T = 112.5$ mg/dl. In this case, we have $T_{\text{min}}+\alpha<T<T_{\text{max}}-\alpha$ for some $\alpha>0$ (e.g. $\alpha=1$, $\alpha=5$,$\ldots$, it does not matter). As we describe in Section 2, the sole purpose of the term $\alpha > 0$ is to emphasize that the target level $T$ **should not be equal** to the safety thresholds ($T\neq T_{\text{min}}$ and $T\neq T_{\text{max}}$). This mild condition is included for completeness and it is used in the proof of Theorem 3. Most importantly, $\alpha$, $T$, $T_{\text{min}}$, and $T_{\text{max}}$ are not hyper-parameters that needs to be tuned.
>
> The clinician can choose the $T$, $T_{\text{min}}$, and $T_{\text{max}}$ values differently for each patient based on the treatment goals. ESCADA learns to make insulin dose recommendations ($d_n$) for different meal events ($z_n$) to achieve the target blood glucose level $T$, while respecting the safety thresholds, $T_{\text{min}}$ and $T_{\text{max}}$, that are determined by the clinician. That is, ESCADA’s goal is not to find some best $T$, $T_{\text{min}},$ and $T_{\text{max}}$ values, but to learn to provide better recommendations for a given set of these values. If these values change, then ESCADA will create the safe dose sets and make its recommendations according to these new values (see how the safe dose sets, $S_n(z_n)$, are formed and the recommendations, $d_n$, are made according to $T$, $T_{\text{min}}$, and $T_{\text{max}}$ in Algorithm 1).
>
> We also kindly ask the reviewer to see **Appendix B - Additional Experimental Results** in the **revised version of the supplementary material**.
>
> ## Answer to Weakness 2
>
> We mention in the **Consistency** part of Section 5 that calculating the posterior mean and covariance functions of the Gaussian process (GP) has $O(n^3)$ time and $O(n^2)$ memory complexities where $n$ is the number of observations. When $n$ gets larger, one can resort to the Sparse Variational GP (SVGP) models that offer a low rank $O(m^2 n)$ approximation of the GP posterior where $m$ is the constant number of so-called *inducing variables* that summarize the training data (see below **Titsias, 2009**). ESCADA uses the posterior mean and covariance functions of the GP. If we denote the cardinality of the dose set used in the experiments by $D$, it has $O(D^2)$ time (for constructing the safe set, see eqn. (3)) and $O(D)$ memory complexities. STS samples from the GP posterior at each round, which has an $O(D^3)$ time complexity due to the Cholesky decomposition, and its memory complexity is $O(D)$. Efficient approximate algorithms such as the *decoupled sampling* (see below **Wilson et al, 2020**) use SVGP posteriors and reduce the sampling cost to $O((m+M)D)$ where $M$ is the constant number of *inducing features*. We will extend our discussions on computational complexities of the algorithms using the extra page.
>
> M.K. Titsias, "Variational learning of inducing variables in sparse Gaussian processes," in *AISTATS*, 2009.
>
> J.T. Wilson et al, "Efficiently sampling functions from Gaussian process posteriors," in *ICML*, 2020.
>
> ## Answer to Question 1
>
> No, $T$ does not depend on $z_n$. As mentioned in the answer to “Weakness 1”, the target level $T$ is determined by the clinician at the beginning. $z_n$ is the context that the learner observes at the beginning of round $n$. In our experiments, $z_n$ is the (fasting blood glucose, carbohydrate intake) tuple that characterizes a “meal event”. After observing the context $z_n$, the learner makes a dose recommendation, $d_n$, so that the objective function will attain the target level, that is, $f(z_n,d_n) = T$. As we explain in Section 2, $f$ is unknown, and we are learning it from online data using a Gaussian process model.
>
> ## Answer to Question 2
>
> To estimate the confidence interval of response at each round $n$ for a dose $d \in \mathcal{D}$, we need $\beta_n$ and $\sigma_{n-1}(z_n,d)$ (Lines 157-158), where the context $z_n$ is given at the beginning of the round. Definition of $\beta_n$ is given in Lemma 1. $\sigma_{n-1}(z_n,d)$ is calculated using the posterior update rules for the GPs (see eqn. (2) in [41]). We mention the GP posterior update but exclude the closed form calculations which are standard in GP literature to avoid clutter (Lines 127-130). We will include them in the final version using the extra page.

---

> ### Author Response · Authors · 2022-08-07
> **Dear Reviewer iS7Z**
>
> We would like to extend our gratitude for your valuable comments once again! We hope that our response (31 Jul) has addressed your concerns, and we would be happy to discuss your further comments after reading the response.

---

> > ### Comment · Reviewer_pdcL · 2022-08-09
> > **thanks**
> >
> > I have no additional comments on this matter, the response is clear and sufficient.

---

> > > ### Author Response · Authors · 2022-08-09
> > > **thank you Reviewer pdcL**
> > >
> > > We are delighted to hear that you were satisfied with our response! We wanted to leave a brief note in case you replied to this post by mistake (instead of the one below your review, titled "Dear Reviewer pdcL"). If not, please forgive us for presuming!
> > >
> > > Best wishes,
> > > Authors

---

### Review · Ethics_Reviewer_EACp · 2022-08-05

**Recommendation:**

With the disclaimer that I am not an expert in IRB approvals, my understanding is that it is likely that the authors are partially right in the sense that this type of human involvement (such as the one the clinician had in this study) would fall under a category of exempt studies for which no IRB review is necessary (for example 2ii or 3ii in https://www.ecfr.gov/on/2018-07-19/title-45/subtitle-A/subchapter-A/part-46#46.104). *However*, even so, this is not a decision authors generally make on their own, there is still an exemption form to be filled (with the university/organization they are affiliated with) and approved (e.g. see wiki page on Institutional Review Board: “Generally, human research ethics guidelines require that decisions about exemption are made by an IRB representative, not by the investigators themselves”).

(As a side comment, the authors mention in their rebuttal that they will acknowledge the clinician in the final version but my understanding according to the exemption rules for IRBs is that it might be best that the clinician remains anonymous.)

I think it is important for the authors to clarify: did the IRB review the conditions of the study and declared that no ethics approval or consent forms were required? My understanding based on the authors’ rebuttal comment is that they made this assessment on their own, in which case I believe it is unlikely that this issue can be overcome at this stage. I might be missing other rules that state that this type of study does not even require an exemption form to be filled, but the authors have not given a concrete justification (with references) of why this might be the case.

I would suggest deferring the decision for this paper to an IRB expert.

**Ethical Issues:**

Yes

**Ethics Review:**

This paper proposes a system that learns how to make dose recommendations that are “safe” in the sense that a physiological variable is kept in the appropriate range. To evaluate the success of the system, a dataset of “fake”/simulated patient events is used, so it is clear that no IRB approval is needed in this case — there are no human patient data involved. However, in one of the experiments, the performance of the system is being compared with that of a human clinician.
By doing this, the clinician technically participated in a study/survey, in which they were asked to give information (their opinion) about these simulated/hypothetical scenarios, and it might be the case that an IRB approval is needed.

---

### Comment · Area_Chair_oXiy · 2022-08-03
**Please start author-reviewer discussion**

Hi authors and reviewers,

The discussion phase has begun. Please read the other reviews and the author's response (if the authors choose to submit one) and start discussing them with the other reviewers, the authors, and myself.

Note that by default, the authors can see the discussions posted by the reviewers (and vice versa). Please use the "Readers" field to adjust the audience of your post if so wished.

*Our goal is to contribute to the discussion to reach a consensus on each paper*. There is only one week for the discussion (until **August 9**). So please do not wait and start the discussion immediately. Thank you very much.

Best,\
The AC

---

### Meta-Review · Area_Chair_oXiy · 2022-08-26

**Recommendation:** Accept
**Confidence:** Certain

**Metareview:**

I have read all comments and responses carefully.

The reviewers recognized that the problem was a challenging one, and that the paper provides both practical and novel tools and theoretical analysis. However, the reviewers pointed to the lack of numerical studies in the paper (for example, more details about the human clinicians and the patients). That being said, the authors have addressed most constructive comments given by reviewers.

Overall, reviewers agree that this is an important and yet underexplored problem and the authors have provided useful contributions. I, therefore, have decided to recommend the acceptance of the paper.

**Award:**

No

---

### Decision · Program_Chairs · 2022-09-14

Accept